# Nanovised Control Flow Attestation

**Raz Ben Yehuda** [1,†] **, Michael Kiperberg** [2,†] **and Nezer Jacob Zaidenberg** [3,*,†]

1   Faculty of Information Technology, University of Jyväskylä, 40014 Jyväskylä, Finland; raziebe@gmail.com
2   Department of Software Engineering, Shamoon College of Engineering, Beer-Sheva 8410802, Israel;
    michaki1@sce.ac.il
3   Faculty of Computer Science, College of Management Academic Studies, Rishon LeZion 7579806, Israel
*   Correspondence: scipio@scipio.org
†   These authors contributed equally to this work.

**Abstract:** This paper presents an improvement of control flow attestation (C-FLAT) for Linux. C-FLAT is a control attestation system for embedded devices. It was implemented as a software executing in ARM's TrustZone on bare-metal devices. We extend the design and implementation of C-FLAT through the use of a type 2 Nanovisor in the Linux operating system. We call our improved system "C-FLAT Linux". Compared to the original C-FLAT, C-FLAT Linux reduces processing overheads and is able to detect the SlowLoris attack. We describe the architecture of C-FLAT Linux and provide extensive measurements of its performance in benchmarks and real-world scenarios. In addition, we demonstrate the detection of the SlowLoris attack on the Apache web server.

**Keywords:** hypervisor; ARM; Linux; control flow; SlowLoris; TrustZone

## 1. Introduction

C-FLAT by Abera et al. [1] is a technique for attesting an application's control flow on an embedded device. C-FLAT is a dynamic analysis tool. It complements static attestation by capturing the program's runtime behavior and verifies the exact sequence of executed instructions, including branches and function returns. It allows the verifier to trace the program's flow control to determine whether the application's control flow was compromised. Combined with static attestation, C-FLAT can precisely attest embedded software execution.

Originally, C-FLAT design allows attestation for simple systems. However, C-FLAT does not support threads, processes or operating systems. Therefore, most industrial systems are too complex for C-FLAT. Complex systems usually require multi-processing, multi-threading, inheritance, or function pointers, etc. These features are available on a General Purpose Operating System (GPOS); C-FLAT does not support GPOS.

Furthermore, C-FLAT runs on top of TrustZone. TrustZone programming requires high expertise, as well as access to the boot loader code. Such access is usually available only to the SoM (System On a Module) vendor. Here, we provide a similar but more straightforward approach using our a dedicated Nanovisor [2]. In this paper, we extend C-FLAT and eliminate the following limitations:

- Single threads—C-FLAT is available only for a single thread.
- Single Process— C-FLAT is available only for a single process.
- Multi-core—C-FLAT is utilized only for a single process.
- TrustZone access—C-FLAT requires access to TrustZone [3,4], which may not be available in many industrial cases.

The paper's main contribution is adapting C-FLAT for complex applications in a GPOS (Linux). We demonstrate how our system can detect real exploits, such as SlowLoris, that affect production systems and handles a real test case (CVE-2019-9210).

We record the control flow path and send continuous sub-sequences to an attestation server. Furthermore, as a result of using Linux, the attestation server may execute locally. Local execution reduces the risk of transmitting the data over the network and allows for better performance.

Additionally, as we use C-FLAT over a complex general-purpose application, we must carefully approach the performance penalty. For this reason, we offer a novel approach to mitigate it. We run the instrumented section alternately. For example, execute this code section once a second.

Our contribution provides software-only CFI for ARM for software running in a GPOS. We perform the path monitoring in TEE or HYP mode. We provide a context-sensitive CFI and is available for 32 bit applications and is useful on Android Phones. Our Nanovisor software can run on EL2 (HYP mode) or TrustZone. Since it can track suspicious paths, it helps in detecting DOS attacks.

## 2. Related Work

Research in CFI is vast and is available in the hardware and the software. ARMv8.3-PAuth [3] adds the Pointer Authentication Code (PAC) feature to the ARM processors. PAC validates whether the target of an indirect branch is valid (Figure 1). PAC can be used for construction of CFI systems. Unfortunately, currently ARMv8.3 processors are not available, thus not allowing it to be embedded in our solution. In addition, the authentication of PAC is performed in EL0, while our system executes in EL2. Clang CFI [5] is designed to detect schemes of undefined behavior in C++ programs. These schemes have been optimized for performance. The schemes rely on LTO (Link Time Optimization). For better efficiency, the program must be structured such that certain object files are compiled with CFI enabled. The schemes focus on casting between types, incorrect virtual calls and indirect virtual calls. It can be interesting to perform Clang's CFI in our Nanovisor.

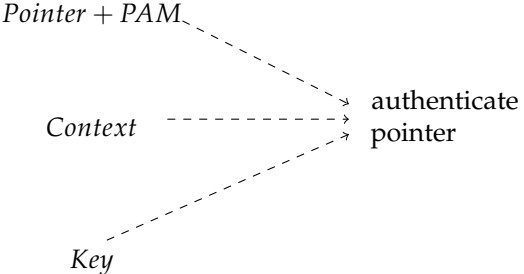

**Figure 1.** PAC.

There are several control flow attestation systems similar to C-FLAT, all having similar limitations. Lo-Fat [6] is similar system for RISC-V architecture. A similar system for the bare metal system was proposed in [7]. Another system for remote control flow attestation of embedded devices was proposed by Liu et al. [8]. Their system is implemented in ARM's TrustZone. ReCFA [9] describes several improvements to the attestation scheme. In particular, ReCFA requires only the binary and not the source code form of the program. In addition, the control flow information is condensed in multiple way, including compression, in order to allow faster and cheaper delivery of this information to the verifier. TinyCFA [10] provides control flow attestation for low-end MCUs, such as ATmega32. Hu et al. proposed [11] to apply the control flow attestation mechanism only to potentially vulnerable parts of the program. Our advantage against all of these is the ability to support GPOS and normal multi-process and multi-threaded applications.

Kernel CFI (kCFI) [12] demonstrates CFI for the Linux kernel. It combines static analysis at the source and binary levels and creates a restrictive CFI policy. Compared to other kernel CFI approaches, it achieves a small overhead, approximately 2%, and supports dynamic module insertion. kFCI is compiler-based technology and, therefore, does not rely on any runtime supervisor or routine, avoiding overheads.

MoCFI [13] (Mobile CFI) is a CFI framework intended for ARM7. It instruments the binary on the fly and offers protection from code injection attacks and code reuse attacks (such as return-to-libc). Unlike C-FLAT, the attestation is performed in a non TEE environment, and the binary analysis is performed offline. However, it is only fair to say that it is possible to migrate the runtime enforcement into our Nanovisor.

PathArmour [14] is a context-sensitive CFI (CCFI) for the Intel platform. Its uniqueness is its ability to detect wrong code path flows; for example, if function B can be called from both A and C, but the current context is A to B, but the actual path is C to B, then PathArmour can detect this. PathArmour uses a kernel module to monitor paths. To reduce the overhead, it mainly focuses on system call tracing and relies on x86_64 branch recording features. Unlike C-FLAT, it does not run in ARM, and the path monitoring is not performed in a TEE.

Another technology for ROP attacks is the kBouncer [15]. KBouncer is a binary CFI for the ×86 platforms. It focuses on ROP attacks protection through the use of Intel's LBR (Last Branch Recording). LBR is a set of registers that record the last branches; it is transparent to the running application and, therefore, incurs zero overhead for storing the branches.

In the area of static attestation, we find SWATT (Software-based Attestation Technique) [16] for embedded devices. SWATT examines the memory content for viruses. Another software in this area is Pioneer [17]. In Pioneer, the executable is guaranteed to execute a trusted environment by implementing a root of trust. This dynamic root of trust is instantiated through the verification function, a self-checking function that checksums its instructions. Viper [18] verifies the integrity of peripherals firmware. SMART [19] is a minor modification to a micro-controller that is used to facilitate a dynamic root of trust in remote embedded devices. TrustLite [20] is an FPGA to enforce software modules protection. Another FPGA solution is TyTan [21], a trust anchor for tiny embedded that provides secured task loading, secured IPC and local and remote attestation.

CFI performance and efficiency had been researched heavily in the past years [4,22,23]. CPI (Control Pointer Integrity) [24,25] is a different technique for insuring pointers in the code. It protects from code hijacking but does not provide information about control flow paths. Property-based attestation [26] however, Refs. [27,28] demonstrated some deficiencies, to name a few, it reveals information about the platform's configuration (hardware and software) and the application, usually through an external attesting agent. In addition, all trusted permutations of the trusted configurations must be known beforehand. Additionally, if the configuration changes, this must reflect a verifier. Software attestation calculates a hash over the program's code, and the correctness also relies on that the verifier responds in time. This real-time requirement may not be possible in a busy network.

C-FLAT and our approach to attestation is dynamic analysis. Using hypervisors for dynamic security of kernels is described in Secvisor [29]. Secvisor can assure that only whitelisted code executes. However, Secvisor does not protect against return-oriented programming or perform dynamic analysis. Liu et al. [30] describe KSP (Kernel Stack Protection), through the use of a hypervisor. In [30], a return-to-schedule attack is made on some process while this process is not executing and its stack has tampered. The protection model is by shadow page tables. These pages are used to provide different kernel stacks with different access permissions; thus, kernel units have different access permissions when they try to modify other kernel stacks; in addition, Liu et al. [30] offer to use the hypervisor to record important information regarding the process, and, thereby, protecting from some malicious kernel. MOSKG (Multiple Operating Systems Kernel Guard) is aimed to protect from DKOM attacks (Dynamic Kernel Object Manipulation), Yan et al. [31] offers a secure paging mechanism to protect critical kernel data.

HIMA[32] is a similar system for detecting kernel integrity using a hypervisor. We performed a dynamic analysis of kernel code using Lguest in [33]. The method was extended [34] for detecting operating system bugs and kernel vulnerabilities. Lgdb is based on a para-virtualized environment and significant performance penalties. It is limited to a single version of Linux (running on the host) and 32 bit ×86. In contrast, our new

approach is using hardware virtualization with good performance. Our approach is aimed directly at detecting ROP (Return Oriented Programming) in real-time. Our new approach is not limited to Linux of any flavor, provides much better performance, and uses ARM hardware-assisted virtualization.

Another approach to dynamic analysis is the phases approach. On the first (online) step, the entire memory of the inspected system is grabbed [35]. Then, the memory is examined using volatility [36] or AI-based tools to detect anomalies in memory [37]. The problem of live memory forensics is that the comparable memory is large. Therefore, changes occur while the forensic analysis is running, causing anomalies [38]. Unlike our method, independent memory acquisition and analysis are not capable of understanding context and are vulnerable to detection anomalies. In contrast, we trace the system stack and are aware of loading libraries and performing function and system calls.

### 3. Threat Model

C-FLAT does not target specific attacks (ROP, MOSKG, DOS) attacks or similar attacks. Therefore, it is challenging to compare C-FLAT to other technologies. C-FLAT targets vulnerabilities once a zero-day attack takes place. Due to its versatile nature, C-FLAT also records the time of execution and, therefore, stipulates better diagnostics of inconsistencies in the time of execution in the flow of the program.

For this paper, an attacker may attack via the network through DOS attacks, or in the case of CFI attacks, the attacker may change the program's memory by exploiting some vulnerability [39]. In this paper, a CFI attack refers to an attack on a native binary. For DOS attacks, we assume that the HTTP request bypassed any network firewall if one exists.

### 4. Background

C-FLAT is a dynamic attestation and complements static attestation. It measures the program's execution path at the opcode level by capturing its runtime behavior. Figure 2 (taken from the original C-FLAT paper) presents the C-FLAT. *Prv* is a prover, and *Ver* is the verifier. The Prover usually runs on a resource-limited device, and, therefore, cannot do the verification while in execution. Both the Verifier and the Prover must have access to the program's binary at any time.

First step ① is that *Ver* generates a control-flow-graph offline. It also measures each possible control-flow path using function *H*, and saves the results ②. In the original C-FLAT, *Ver* stores and generates CFG's results in *Ver*, because *Prv* is a low resources' computer. In C-FLAT Linux the CFG is stored in *Ver* itself. Reading the CFG is required only once per program. We do it on the process start-up, so it does not endanger the performance. In ③, *Ver* challenges *Prv*. Then, *Prv* starts executing the program ④, while our trusted software computes the program's path ⑤. In the original C-FLAT, code running in TrustZone performed this computation. In our system code, running in hypervisor mode performs the calculation. Lastly, *Prv* generates the attestation report $r = SigK(Auth, c)$, computed as a digital signature over the challenge c and Auth using a key K known only to the measurement engine. Lastly, *Prv* transmits r to *Ver* ⑥ to validate it ⑦.

C-FLAT model is designed for small embedded devices, and therefore does not fit to a general-purpose OS on a general-purpose processor which is common today in many embedded setups. C-FLAT Linux is suitable for these setups.

In addition to the above, for C-FLAT to trace the program's flow, it is essential to intervene in the program binary (Original). For this, we replace the program's branches and returns-from-function. For example, in Figure 3, the "br x0" opcode is replaced by "br hook_b" opcode (Instrumented). Hook_b is an address of a procedure that searches for the address of the original branch opcode, and once found (Trampoline), it sends (via the SMC command) it to computation in the TrustZone. The computation records the address and returns to perform the original branch command.

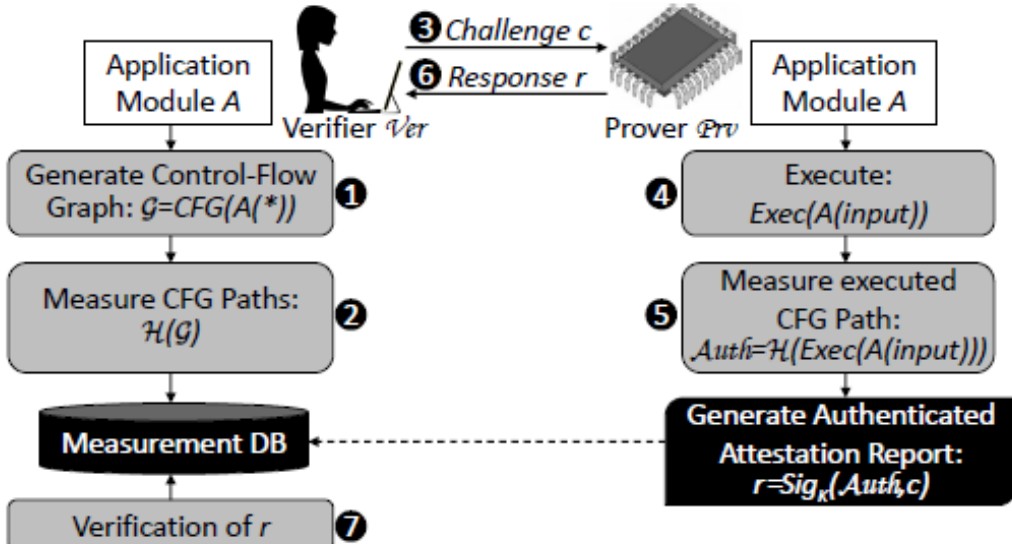

**Figure 2.** C-FLAT original system model ©Abera.

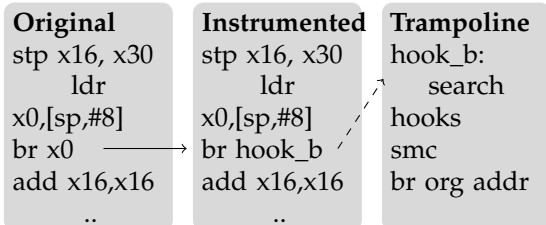

**Figure 3.** C−FLAT Instrumented.

## 5. Control Flow Attestation for Linux

Our Nanovisor implementation (Figure 4) replaces the TrustZone. C-FLAT Linux instruments the ELF [40] (Executable Linking Format) such as in [1]. The instrumentation includes replacing the binary opcodes for the various "branch" commands with a code that branches to the trampoline. It also requires caching the original branch targets.

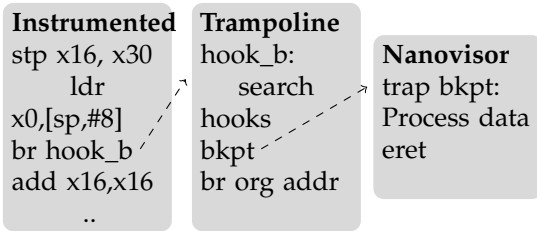

**Figure 4.** A C−FLAT Program in Linux.

Figures 3 and 4 depict the main difference in the infrastructures. Our Nanovisor provides a trusted execution environment (TEE) through a fast Nanovisor RPC [2]. The Nanovisor performs the computation and aggregation of addresses generated by the program's flow.

Figure 5 depicts that as the instrumented program $P$ runs, it generates input to Nanovisor $Q$. $Q$ collects the input, and at some point, the collected data are encrypted and passed to the attestation server. The attestation server deciphers the data and verifies that the control path does not have any abnormalities. The attestation server handles each branch instruction or batches of instructions.

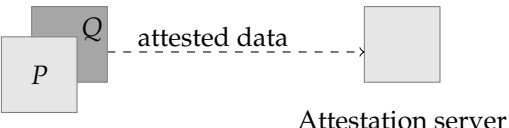

**Figure 5.** C-FLAT architecture.

Program *Q* (Figure 5) is a background process that serves the Nanovisor requests from the protected program. Parts of *Q* execute in HYP mode through the use of our Nanovisor RPC [2]. *Q* loads when the computer boots and waits in the background. C-FLAT does not start any threads or agents, and its overhead is small. *Q*'s main task is to process the information from the protected process and pass it to the attestation server. The information that the protected program passes *Q* is the Event Type, the Destination address, the Source address and Link Register Value. These values are passed on four registers to increase performance. However, the constant round trip of the traps incurs an overhead. We will demonstrate the overhead in the evaluation section. To improve performance, we decided to mitigate the number of round-trips.

*5.1. Performance*

As we aim to mitigate the traps penalty, we need to minimize the amount of Nanovisor traps (calls). Thus, we trigger off and on the trap code. To do that, we intervene in the protected program execution code while it executes and replace the trap opcode (the BKPT instruction) with the NOP (no operation) opcode. A NOP opcode depends on the compiler output, and, therefore, to modify the opcode in real-time, it is required to know in advance the following

1. The size of the BKPT opcode (16 bit or 32 bit);
2. The position of the opcode in the program's address space;
3. Identify that a protected program runs (or sleeps) without the need to wait for traps.

The ARM compiler may generate opcode in thumb mode (16bit) or regular mode (32bit). For this reason, it is essential to know the BKPT (breakpoint) opcode size and position. Therefore, the Nanovisor records the position (and size) of the first trap generated by the INIT operation (Figure 6). The INIT operation is also used to initialize C-FLAT's background process (*Q*). The page containing the trap code is mapped to the kernel at some early point in the program execution. The user (which may be a program), at his choice, triggers the coding and re-coding of the BRPT/NOP opcodes.

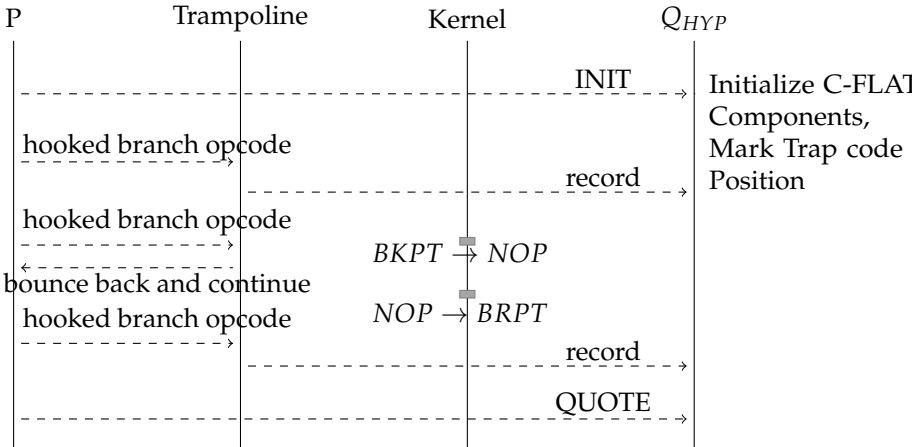

**Figure 6.** C-FLAT Flow.

The trap to the Nanovisor is performed by programming the Nanovisor to trap access to BKPT by setting the mdcr_el2 register (breakpoint control register). This means that any program that triggers a breakpoint enters the Nanovisor. Since there are systems that

require debug infrastructure, only when the protected program executes the trap is set, and after the scheduler switches the process, it is unset. The C-FLAT framework names the protected ELF sections ".attest". Each program invocation triggers a scan for the ".attest" ELF sections. Then, the Nanovisor enables the trap for the protected process. Thus, any other process that triggers a breakpoint does not enter the Nanovisor.

To summarize, when possible, we can set C-FLAT to execute arbitrarily, thus improving performance and reducing processor cycles and heat.

### 5.2. GPOS Considerations

Original C-FLAT runs on an embedded ARM-based system without a general purpose operating system (GPOS). For example, Linux, being a GPOS, allows execution of processes on multi-core architectures, thus enabling processes to migrate from one core to another. Figure 7 demonstrates how multiple C-FLAT threads of the same process run concurrently. This aspect requires our solution to be deployed on all cores.

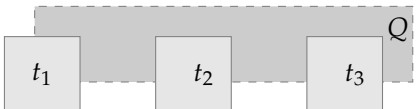

**Figure 7.** C-FLAT multi-core server.

Another facet of GPOS is context switching. Assuming that a protected program P is infected by a virus, and thereby it refrains access to the Nanovisor. When C_FLAT is executing in a non GPOS, it is likely, that most if not all of the time, a single process occupy the processor. Therefore, after some time that C-FLAT did not execute, the system may be infected by a virus. In a GPOS, we cannot make that assumption, as other processes may pre-empt the protected program or that the protected program is not executing. For that, we need to know that a C-FLAT process executes, and when it does, we expect C-FLAT to trigger shortly. For this reason, in each context switch, we also check whether the C-FLAT program enters or leaves the processor and record the time and the traps count.

Multi-threading also challenges C-FLAT because threads are created and destroyed ad-hoc. We, therefore, offer that each thread state is kept in a table entry in the C-FLAT server. Once a thread is destroyed, we mark this entry as free. We use the TLS (thread-local storage) [41] to differentiate between the different threads and processes. The Linux kernel in ARM puts on the tpidr_el0 register the TLS value, which is accessible in the Nanovisor.

Multiple C-FLAT processes are possible. Multi-processes design is similar to the multiple threads design we discussed.

### 5.3. The Hyplet Nanovisor

Here, we describe the Nanovisor technology that we referred to as the hyplet [42,43]. ARM8v-a specifications offer to distinct between user-space addresses and kernel space addresses by the MSB (most significant bits). The user-space addresses of Normal World and the hypervisor use the same format of addresses.

These unique characteristics are what make the hyplet possible. The Nanovisor can execute user-space position-independent code without preparations. Consider the code snippet at Figure 8. The ARM hypervisor can access this code's relative addresses (adrp), stack (sp_el0) etcetera without pre-processing. From the Nanovisor perspective, Figure 8 is a native code. Here, for example, address $0 \times 400,000$ is used both by the Nanovisor and the user.

```
400610: foo:
400614: stp x16, x30, [sp,#−16]!
400618: adrp x16, 0x41161c
40061c: ldr x0, [sp,#8]
400620: add x16, x16, 0xba8
400624: br x17
400628: ret
```

**Figure 8.** A simple hyplet .

So, if we map part of a Linux process code and data to a Nanovisor, it can be executed by it.

To make sure that the program code and data are always accessible and resident, it is essential to disable evacuation of the program's translation table and cache from the processor. Therefore, we chose to constantly accommodate (cache) the code and data in the hypervisor translation registers in EL2 cache and TLB. To map the user-space program, we modified the Linux ARM-KVM [44] mappings infrastructure to map a user-space code with kernel space data.

Figure 9 demonstrates how identical addresses may be mapped to the same virtual addresses in two separate exception levels. The dark shared section is part of EL2 and, therefore, accessible from EL2. However, when executing in EL2, EL1 data is not accessible without previous mapping to EL2. Figure 9 presents the leverage of a Linux process from two exception levels to three.

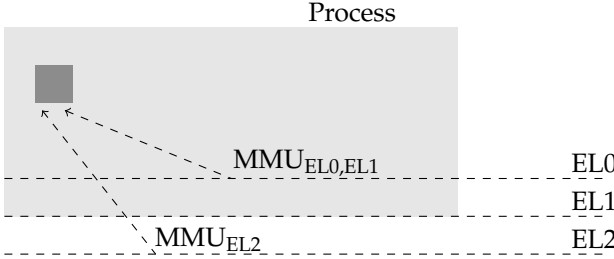

**Figure 9.** Asymmetric dual view. Two exception levels access the same physical frame with the same virtual address of some process. However, the page tables of the two exception levels are not identical.

### 5.3.1. The Hyplet Protection

The natural way of memory mapping is that EL1 is responsible for EL1/EL0 memory tables, and EL2 is responsible for its memory table, in the sense that each privileged exception level accesses its memory tables. However, this would have put the Nanovisor at risk, as it might overwrite or otherwise garble its page tables. As noted earlier, on ARM8v-a, The hypervisor has a single memory address space (unlike TrustZone that has two, for kernel and user). The ARM architecture does not coerce an exception level to control its memory tables. This makes it possible to map the EL2 page table in EL1. Therefore, only EL1 can manipulate the Nanovisor page tables. We refer to this hyplet architecture as a Non-VHE hyplet. Additionally, to further reduce the risk, we offer to run the hyplet in injection mode. Injection mode means that once the hyplet is mapped to EL2, the encapsulating process is removed from the operating system kernel, but its hyplet's pages are not released from the Nanovisor, and the kernel may not re-acquire them. It is similar to any dynamic kernel module insertion.

In processors that support VHE (Virtual Host Extension), EL2 has an additional translation table that would map the kernel address space. In a VHE hyplet, it is possible to execute the hyplet in the user space of EL2 without endangering the hypervisor. A hyplet of a Linux process in $EL0_{EL1}$ (EL0 is EL1 user-space) is mapped to $EL0_{EL2}$ (EL2

user-space). Additionally, the hyplet cannot access EL2 page tables because the table is accessible only in the kernel mode of EL2. VHE resembles TrustZone as it has two distinct address spaces, user and kernel. Operating systems, such as QSEE (Qualcomm Secure Execution Environment) and OP-TEE [45], are accessed through an up-call and execute the user-space in TrustZone. Unfortunately, at the time of writing, only modern ARM boards offer VHE extension (ARMv8.2-a) and therefore, this paper demonstrates benchmarks on older boards.

### 5.3.2. The Hyplet Security

As noted, VHE hardware is not available at the time of this writing, and, as such, we are forced to use software measures to protect the hypervisor. On older ARM boards, it can be argued that a security bug at hypervisor privilege levels may cause greater damages compared to a bug at the user process or kernel levels, thus poising system risk.

The hyplet also escalates privilege levels, from exception level 0 (user mode) or 1 (OS mode) to exception level 2 (hypervisor mode). Since the hyplet executes in EL2, it has access to EL2 and EL1 special registers. For example, the hyplet has access to the level 1 exception vector. Therefore, it can be argued that the hyplet comes with security costs on processors that do not include ARM VHE.

The hyplet uses multiple exception levels and escalates privilege levels. So, it can be argued that using hyplets may damage application security. Against this claim, we have the following arguments.

We claim that this risk is superficial and acceptable, for processors without VHE support. Most embedded systems and mobile phones do not include a hypervisor and do not run multiple operating system.

In the case where no hypervisor is installed, code in EL1 (OS) has complete control of the machine. It does not have a lesser access code running in EL2, since no EL2 hypervisor is present. Likewise, code running in EL2 can affect all operating systems running under the hypervisor. Code running in EL1 can only affect the current operating system. When only one OS is running, the two are identical.

Therefore, from the machine standpoint, code running in EL1 when EL2 is not present has similar access privileges to code running in EL2 with only one OS running, as in the hyplet use case.

The hyplet changes the system from a system that includes only EL0 and EL1 to a system that includes EL0, EL1, and EL2. The hyplet system moves a code that was running on EL1 without a hypervisor to EL2 with only one OS. Many real-time implementations move user code from EL0 to EL1. The hyplet moves it to EL2; however, this gains no extra permissions; running rogue code in EL1 with no EL2 is just as dangerous as moving code to EL2 within the hyplet system. Additionally, it is expected that the hyplet would be a signed code; otherwise, the hypervisor would not execute it.

The hypervisor can maintain a key to verify the signature and ensure that the lower privilege level code cannot access the key.

Furthermore, real-time systems may eliminate even user and kernel mode separation for minor performance gains. We argue that escalating privileges for real performance and real-time capabilities is acceptable on older hardware without VHE where hyplets might consist of a security risk. On current ARM architecture with VHE support, the hyplet do not add extra risk.

### 5.3.3. Static Analysis to Eliminate Security Concerns

Most memory (including EL1 and EL2 MMUs and the hypervisor page tables) is not mapped to the hypervisor. The non-sensitive part of the calling process memory is mapped to EL2. The hyplet does not map (and, thus, has no access to) kernel-space code or data. Thus, the hyplet does not pose a threat of unintentional corrupting kernel's data or any other user process unless additional memory is mapped or EL1 registers are accessed.

Thus, it is sufficient to detect and prevent access to EL1 and EL2 registers to prevent rogue code affecting the OS memory from the hypervisor. We coded a static analyzer that prevents access to EL1 and EL2 registers and filters any special commands.

We borrowed this idea from eBPF [46]. The code analyzer scans the hyplet opcodes and checks that are no references to any black-listed registers or special commands. Except for the clock register and general-purpose registers, any other registers are not allowed. The hyplet framework prevents new mappings after the hyplet was scanned to prevent malicious code insertions. Another threat is the possibility of the insertion of a data pointer as its execution code (In the case of SIGBUS of SEGFAULT, the hyplet would abort, and the process terminates). To prevent this, we check that the hyplet's function pointer, when set, is in the executable section of the program.

Furthermore, the ARM architecture features the TrustZone mode that can monitor EL1 and EL2. The TrustZone may be configured to trap illegal access attempts to special registers and prevent any malicious tampering of these registers.

The Nanovisor is 768 lines of code. It includes the interrupt vector (300 lines) and the hyplet's user-kernel interface. It is part of the Linux kernel and, therefore, open-source. Its tiny size eases code analysis and its protection (ex. Hypersafe [47]). As noted, we reused Linux's KVM infrastructure. KVM is well-debugged, and thus it reduces risks of errors; Additionally, future development in the area of virtualization in Linux may be adapted to the hyplet.

Control flow attestation demands a facility for static remote attestation capable of attesting the instrumented code. Otherwise, the control flow might be easily spoofed by an adversary that adds or removes instrumentation or traps. Therefore, we suggest encrypting the ensuring function, as suggested in Ben Yehuda et al. [2]. Ben Yehuda et al. [2] presented a technique to sign pieces of code (functions and data) digitally through the use of the hyplet.

## 6. Evaluation

This section provides the evaluation results of our system. The evaluation was perform on Raspberry PI3 boards, whose specification is provided in Table 1. This section describes several tests that were executed to assess the performance of C-FLAT Linux and its security. Each test is designed to reveal a certain aspect of C-FLAT Linux performance and security. The first test estimates the transition latency to the Nanovisor. The second test builds on this result and dissects the overall latency to its components. The third test demonstrates the performance of C-FLAT Linux in a real-world scenario. The fourth test demonstrates the performance of C-FLAT Linux during the execution of the Apache2 web server. Finally, in the last test, we demonstrate the security aspect of C-FLAT Linux.

**Table 1.** PI3 specifications.

| Soc | Broadcom BCM2837 |
| --- | --- |
| CPU | 4 cores, ARM Cortex A53, 1.2 GHz, (clocked to 700 MHz) |
| RAM | 1GB LPDDR2 (900 MHz) |
| Clock | 19.2 Mhz |

*6.1. Test 1: Transition Latency*

In order to estimate the performance overhead due to transition from and to the Nanovisor, we set a trap on the BRK opcode from the trampoline to the attestation server. Table 2 presents the time to move from user space to the Nanovisor.

**Table 2.** Transition latency.

| Measure | BRK Trap |
|---------|----------|
| Avg | 92 ns |
| StdDev | 41 ns |
| Max | 156 ns |
| Min | 52 ns |

*6.2. Test 2: Latency Dissection*

In order to measure the overhead of C-FLAT Linux in its two modes of operation, we have written and executed a CPU-intensive program depicted in Figure 10. The functions `do_odd` and `do_parity` multiply their two arguments and return the result. The function `foo` runs a 100,000 iterations loop that alternately invokes the `do_odd` and `do_parity` functions. The program was compiled and executed 10 times in each of the following configurations:

- Without C-FLAT Linux—to establish the baseline performance;
- With C-FLAT Linux with its BKPT opcodes replaced by the NOP opcode—to measure the degradation associated with the presence of C-FLAT Linux;
- With C-FLAT Linux and full instrumentation—to measure the maximum performance degradation.

Table 3 presents the execution times of the compiled program in each configuration.

```
extern void do_odd(int x,int y);
extern void do_parity(int x,int y);

void foo(int loops)
{
for(int i = 0; i < loops; i++){
if ( i % 2 )
do_odd(i,loops);
else
do_parity(i, loops);
}
}
int main() {
foo(100000);
}
```

**Figure 10.** Measurement program for C-FLAT Linux.

**Table 3.** Program execution times in $\mu$-seconds.

| Test | Avg | Max | Min | Std Dev |
|------|-----|-----|-----|---------|
| No C-FLAT | 5188 | 5823 | 5055 | 241 |
| C-FLAT with NOPs | 37,097 | 37,237 | 36,979 | 92 |
| C-FLAT with BKPT | 148,056 | 154,086 | 146,994 | 2174 |

In order to calculate the attestation latency, we use the following observations:

- In each operation, the program jumps to the trampoline twice: in the condition of the *for* and in the condition of the *if*;
- The number of iterations if 100,000;
- The number of trampoline invocations is 200,000;

- According to Table 2, a single entry to the Nanovisor takes 92 nanoseconds; 200,000 entries will take 18,400 µs.

  Therefore, the attestation latency can be expressed as follows:

$$148{,}056 - 5188 - 37{,}097 - 18{,}400 = 87{,}371 \text{ µs}$$

where:

- 148,056 represents the total execution time in the "C-FLAT with BKPT" configuration;
- 5188 the total execution time in the "No C-FLAT" configuration;
- 37,097 the total execution time in the "C-FLAT with NOPs" configuration;
- 18,400 represents the time required to enter the Nanovisor.

The attestation itself is responsible for $\frac{87}{148} = 58\%$ of the total execution time.

### 6.3. Test 3: Real-World Performance

In this test we execute the AdvanceCOMP 2.1 [48] to compress PNG images. Due to its documented vulnerability (CVE-2019-9210), some inputs cause the program to crash. We have executed the program 10 times with legal and erroneous inputs and measured its average execution times. Table 4 presents the obtained results.

**Table 4.** AdvanceCOMP 2.1 performance.

| Input | Test | Avg | Max | Min | Std Dev |
|-------|------|-----|-----|-----|---------|
| Legal | No C-FLAT | 1628 | 1688 | 1606 | 31.8 |
| Legal | C-FLAT with BKPT | 2655 | 2795 | 2625 | 52 |
| Erroneous | No C-FLAT | 21.7 | 23 | 21 | 0.67 |
| Erroneous | C-FLAT with BKPT | 113 | 119 | 111 | 3 |

From Table 4 it is evident that C-FLAT incurs an overhead. In the erroneous input, the overhead is five times larger, while in the good input, it is two times larger. The reason is that the good input is much bigger (1/2 MB on average) than the erroneous input (200 bytes), and, therefore, less I/O activity is involved. The standard deviation of the erroneous input is 4.5 times bigger in the C-FLAT mode, while it is only 1.6 times bigger for the legal input. We believe that this can be explained by the additional, time-consuming activities performed by the operating system due to the program's abnormal termination in case of an erroneous input.

### 6.4. Test 4: Web Server Performance

We used C-FLAT Linux to detect the SlowLoris Denial of Service (DoS) attack [49,50] on the Apache2 [51] web server (version 2.2.3). SlowLoris does not necessarily alters the dataflow path, but rather it changes the rate at which certain operations are performed. The Apache2 web server is a multi-process program. After its initialization, it spawns six processes. During an active attack this number raises to 240. We use the Apache web server to demonstrate that C-FLAT Linux is capable of analyzing control flow of multi-process programs. Each Apache2 process may execute on any processor, and it is being invoked arbitrarily. To accommodate this behavior we deployed the C-FLAT Linux attestation on all the processors as in Figure 7.

For our test, we have selected the following Apache2 routines for attestation, which resulted in 46 hooks, covering part of the program's control flow:

- `unixd_accept;`
- `default_handler;`
- `core_create_req;`
- `core_create_conn;`
- `core_pre_connection.`

After instrumenting the Apache2 web server, we used SlowHttpTest [52] (version 1.6) to test the server in two configurations: "No C-FLAT" and "C-FLAT with BKPT". During both tests, the web server was running on Raspberry PI. The exact command line is given in Figure 11. The command should be interpreted as follows: (-c) initiate 500 connections, (-g) generate CSV files, (-H) use SlowLoris mode, i.e., send unfinished HTTP requests, in a rate (-r) of 200 connections per second, (-t) use the HTTP verb GET, (-x) follow up 24 bytes of data for SlowLoris and POST tests, and probe a connection for 2 s.

```
slowhttptest -c 500 -H -g -o ./outfile
-i 10 -r 200 -t GET -u http://192.168.1.13
-x 24 -p 2 -l 20
```

**Figure 11.** SlowHttpTest command line

Figures 12 and 13 demonstrate that C-FLAT Linux does not affect the performance of the web server under the SlowHttpTest benchmarking tool.

The PI processor did not show any difference in its consumption between the two runs. The Apache2 consumed on average, 2% of the CPU time.

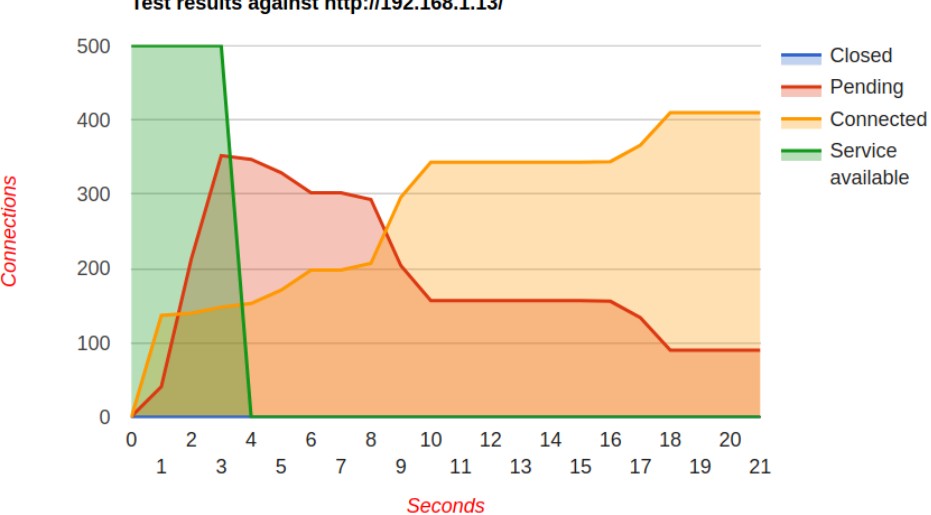

| Test parameters | |
| --- | --- |
| Test type | SLOW HEADERS |
| Number of connections | 500 |
| Verb | GET |
| Content-Length header value | 4096 |
| Extra data max length | 52 |
| Interval between follow up data | 10 seconds |
| Connections per seconds | 200 |
| Timeout for probe connection | 2 |
| Target test duration | 20 seconds |
| Using proxy | no proxy |

**Figure 12.** SlowLoris protected.

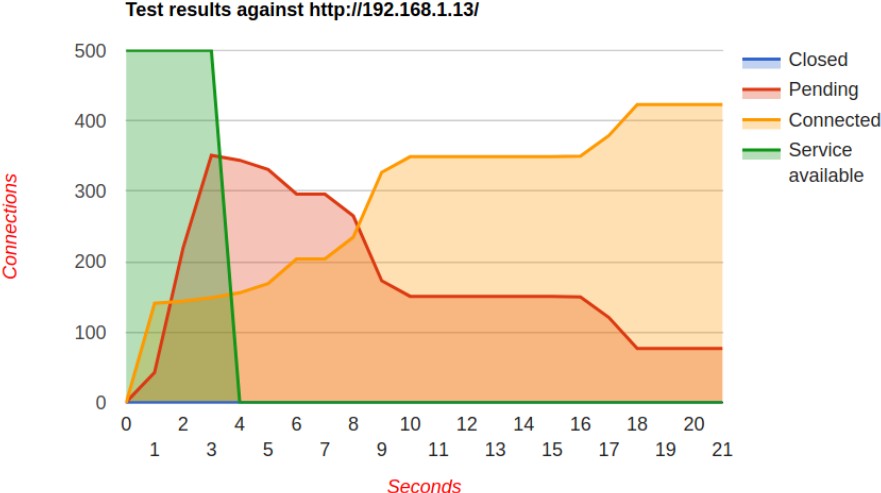

| Test parameters | |
|---|---|
| **Test type** | SLOW HEADERS |
| **Number of connections** | 500 |
| **Verb** | GET |
| **Content-Length header value** | 4096 |
| **Extra data max length** | 52 |
| **Interval between follow up data** | 10 seconds |
| **Connections per seconds** | 200 |
| **Timeout for probe connection** | 2 |
| **Target test duration** | 20 seconds |
| **Using proxy** | no proxy |

**Figure 13.** SlowLoris unprotected.

*6.5. Test 5: Security*

To detect the SlowLoris attack we recorded a legal access flow of the Firefox web browser (version 75.0 64bit), Chrome web browser (version 80.0.3987.132 64-bit) and wget (version 1.19.4). We accessed the Apache2 remotely over wireless from an Ubuntu machine.

Table 5 demonstrates the program flow between INIT and QUOTE in the three runs. The hooked opcodes here are the pop operation (marked p) the branch operation (marked b). Pop is the hook for the "pop of frame pointer and returns address off stack" instruction, and "branch" is the hook for the "branch" instruction.

**Table 5.** Browser vs. SlowLoris.

| SlowLoris | b | b | b | b | p | b | b | p | b |
|---|---|---|---|---|---|---|---|---|---|
| Browser | b | b | b | b | b | b | b | | |
| wget | b | b | b | b | b | | | | |

From Table 5 it is evident that the number of operations is different, and the program flow is different. The INIT to QUOTE in the SlowLoris run belongs to a single process and is a repeated snippet of 4600 operations recorded. Therefore, the SlowLoris attack is detectable by C-FLAT Linux.

## 7. Discussion

The first notable advantage of C-FLAT Linux is its low ($\approx$92 ns) overhead for transitions to the Nanovisor, as is evident from Table 2. For comparison, as was shown previously [2], the latency of transitions to the operating system is $\approx$500 ns. However, despite such low transitions costs, their frequency renders the overall performance inapplicable

to CPU-intensive programs. As shown in Table 3, the instrumentation alone degrades the performance by a factor of 7; transitions to the Nanovisor adds a factor of 4. From the overall performance degradation $\approx 58\%$ are due to the attestation algorithm. We conclude that C-FLAT and, therefore, C-FLAT Linux are inappropriate for full control flow attestation of CPU-intensive programs. However, C-FLAT Linux achieves negligible performance overhead in IO-intensive programs, as demonstrated by Figures 12 and 13.

## 8. Summary

### 8.1. Future Work

The trampoline is 40% of the total penalty. Therefore, to improve the mitigation, we intend to replace the trampoline code of the running process by the NOP opcode.

We intend to add to C-FLAT the ability to analyze and instrument a process dynamically. We intend to inject hooks into a running program. As the program loads, we examine in each context switch where the program counter is, record its position, and at some point, inject hooks to the most accessed code in the program.

### 8.2. Conclusions

It is only fair to say that other technologies, such as MOSKG [31] or KSP [30] provide only a few percentage overheads for micro-benchmarks. However, as noted, these technologies do not offer a versatile solution as C-FLAT Linux.

We conclude that though this paper lessons the performance penalty, there is still work to do in this area. Thus, C-FLAT Linux may be used in non-CPU intensive applications, such as web servers or command-line utilities, and it may also be used in an I/O intensive programs.

**Author Contributions:** All authors had equal contribution. Conceptualization R.B.Y. and N.J.Z.; Methodology, M.K. and N.J.Z.; Software R.B.Y.; Validation, R.B.Y. and N.J.Z.; formal analysis M.K.; Investigation, R.B.Y.; resources N.J.Z. and M.K. writing—original draft preparation, R.B.Y.; writing—review and editing, M.K. and N.J.Z.; visualization, R.B.Y.; supervision, N.J.Z.; project administration, N.J.Z. funding acquisition, M.K. and N.J.Z. All authors have read and agreed to the published version of the manuscript.

**Funding:** This research received no external funding.

**Institutional Review Board Statement:** Not applicable.

**Informed Consent Statement:** Not applicable.

**Data Availability Statement:** Please contact Raz Ben Yehuda for sources and exact benchmark data.

**Conflicts of Interest:** The authors declare no conflicts of interest.

## Abbreviations

Glossary:

| | |
|---|---|
| VHE | Virtual Host Extension |
| EL | Exception Level |
| BKPT | Breakpoint |
| TEE | Trusted Execution Environment |
| RPC | Remote Procedure Call |
| ISR | Interrupt Service Routine |
| ELF | Executable and Linking Format |
| SMC | System Monitor Call |
| SVC | System Supervisor Call |
| *Ver* | Verifier |
| *Prv* | Prover |
| MOSKG | Multiple Operating Systems Kernel Guard |
| DKOM | Dynamic Kernel Object Manipulation |
| ROP | Return Oriented Programming |

| KSP | Kernel Stack Protect |
| DOS | Denial Of Service |
| CFI | Control Flow Inspection |
| CPI | Control Pointer Integrity |
| PAC | Pointer Authentication Code |
| NOP | No Operation |
| TLS | Thread Local Storage |
| QSEE | Qualcomm Secure Execution Environment |
| OP-TEE | Open Portable Trusted Execution Environment |

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
