# Peer review of "Nanovised Control Flow Attestation"

_applsci, doi:10.3390/app12052669_

Round 1

Reviewer 1 Report

The article presents an improvement of control flow attestation (C-FLAT) for Linux.

Presented problem and obtained results are interesting. However, the article has some weaknesses and points worth explaining.

- In introduction authors should mention other methods for attesting an application’s control flow on an embedded device.

- Section 6 authors should extend discussion of obtained results (tables and figures).
The presented conclusions are not obvious.

- Additionally, figures are too small and illegible.

- I suggest to extend the conclusions section.
Section Conclusions ought not be divided into subsections.

Reviewer 2 Report

The paper presents an technical improvement of a control flow attestation. While the authors have a valuable contribution, there are several problems with its presentation:

(1) the content is not matching very well the "applied science" topics, being highly specific for computer security specialists

(2) the contributions are mainly technical, with code snapshots un-specific for a scientific paper

(3) the presentation style  is not appropriate:

  • It is based on based on short and unclear sentences. E.g. "not available at the time of writing. Also, the authentication is done in " [N.R. why also?], "there are additional considerations. First of which is Multi-core."[N.R. Multi-core is a consideration?"] "We provide benchmarks and compare our solution to Normal runs. We use Raspberry 368 PI3 to demonstrate." [N.R. Where is the reference to Normal; to demonstrate what?
  • Table 4 with 1 line and no header (should be a figure!) 
  • typos like "Therefor,"

Reviewer 3 Report

Good approach, further engineering activity expected. Revision in presentation style and language would be appreciated.

Author Response

No comments

Round 2

Reviewer 2 Report

The paper has been changed carefully following the request of the reviewers. The new text do not create new concerns.